# Technical note: Interferences of volatile organic compounds (VOC) on methane concentration measurements

Lukas Kohl[1,2], Markku Koskinen[1,2,3], Kaisa Rissanen[2,4], Iikka Haikarainen[1,2], Tatu Polvinen[1,2], Heidi Hellén[3], and Mari Pihlatie[1,2,5]

[1]Department of Agricultural Sciences, University of Helsinki, Helsinki, 00790, Finland
[2]Institute for Atmospheric and Earth System Research / Forest Sciences, Faculty of Agriculture and Forestry, University of Helsinki, Finland
[3]Finnish Meteorological Institute, PO.Box 503, 00101 Helsinki, Finland
[4]Department of Forest Sciences, University of Helsinki, Helsinki, 00790, Finland
[5]ViPS - Viikki Plant Science Center, University of Helsinki, Helsinki, 00790, Finland

**Correspondence:** Lukas Kohl (lukas.kohl@helsinki.fi)

**Abstract.** Studies that quantify plant methane ($CH_4$) emission rely on the accurate measurement of small changes in the mixing ratio of $CH_4$ that coincide with much larger changes in the mixing ratio of volatile organic compounds (VOCs). Here, we assessed if 11 commonly occurring VOCs (e.g., methanol, $\alpha$- and $\beta$-pinene, $\Delta$3-carene) interfered with the quantitation of $CH_4$ by five laser absorption spectroscopy and Fourier-transformed infrared spectroscopy (FTIR) based $CH_4$ analysers, and quantified the interference of seven compounds on three instruments. Our results showed minimal interference with laser based analysers, and underlined the importance of identifying and compensating for interferences with FTIR instruments. When VOCs were not included in the spectral library, they exerted a strong bias on FTIR-based instruments (64 - 1800 ppbv apparent $CH_4$ / ppmv VOC). Minor (0.7 - 126 ppbv / ppmv) interference with FTIR based measurements were also detected when the spectrum of the interfering VOC was included in the library. In contrast, we detected only minor (<20 ppbv / ppmv) and transient (<1 minute) VOC interferences on laser absorption spectroscopy based analysers. Overall, our results demonstrate that VOC interferences have only minor effects on $CH_4$ flux measurements in soil chambers, but may severely impact stem and shoot flux measurements. Laser absorption based instruments are better suited to for quantifying $CH_4$ fluxes from plant leaves and stems than FTIR based instruments, significant interferences in shoot chamber measurements could not be excluded for any of the tested instruments. Our results furthermore showed that FTIR can precisely quantify VOC mixing ratios , and could therefore provide a method complementary to proton-transfer-reaction mass spectrometry (PTR-MS).

## 1 Introduction

Gas analysers based on infrared spectroscopy are increasingly used to study fluxes of $CH_4$ and other trace gases in natural and anthropogenic ecosystems (e.g. Zellweger et al., 2016; Etiope, 2015; Rapson and Dacres, 2014). Laser absorption spectroscopy based on cavity ring-down spectroscopy (CRDS) or off-axis integrated cavity output spectroscopy (OA-ICOS) is currently considered state of the art by international flux stations networks (Franz et al., 2018). These analysers quantify trace gas mixing ratio through absorption at one specific wavelength. Fourier-transformed infrared spectroscopy (FTIR) is another approach to

measure trace gas fluxes that is gaining popularity because of lower costs, easier field portability, and great versatility with regards to target compounds analytes (e.g. Warlo et al., 2018; Teutscherova et al., 2019; Kandel et al., 2018; Jurasinski et al., 2019). FTIR based analysers measure a complete infrared absorption spectrum, and then quantify the mixing ratio of trace gases through spectral deconvolution using reference spectra for a number of potentially present gases. The capabilities and limitations of both instrument types remain subject of ongoing research. In particular, the potential for biased measurements due to spectral interference with other gases still needs to be established for various environments and applications (e.g. Rella et al., 2015; Assan et al., 2017; Zhao et al., 2012; Zellweger et al., 2016).

Plants were recently identified as an important component of the natural cycles of $CH_4$ (Keppler et al., 2006; Nisbet et al., 2009; Carmichael et al., 2014). This has led to an increased interest in the role of trees in the $CH_4$ exchange of forests (e.g. Pangala et al., 2017, 2015; Machacova et al., 2016; Pitz et al., 2018; Pitz and Megonigal, 2017). Such studies require precise measurements of $CH_4$ emissions from tree stems and shoots, which are typically conducted using the static chamber method where part of a plant (typically shoots or stem areas) places in an enclosure and changes in the mixing ratio of $CH_4$ over time are monitored (Covey and Megonigal, 2019). This monitoring of $CH_4$ mixing ratios was traditionally conducted by collecting chamber air samples at different time points, which were then analysed by gas chromatography (e.g. Machacova et al., 2016). More recently, portable analysers based on CRDS, OA-ICOS or FTIR are increasingly used to measure chamber air $CH_4$ mixing ratios directly in the field (Warner et al., 2017; Pitz and Megonigal, 2017; Pitz et al., 2018). These novel methods have facilitated easier, faster, and more precise measurements of $CH_4$ fluxes, but have also increased vulnerability towards mismeasurements due to spectral interferences. This is especially important in the study $CH_4$ emissions by plants as plants co-emit a complex mixture of volatile organic compounds (VOC) at fluxes 2 to 4 orders of magnitude higher than currently reported $CH_4$ fluxes (Rinne et al., 2002; Simpson et al., 1999; Tarvainen et al., 2005; Machacova et al., 2016; Pangala et al., 2017). The degree to which plant-emitted VOCs interfere with $CH_4$ mixing ratio measurements, however, has so far not been evaluated.

In a recent field campaign, we conducted parallel measurements of tree stem $CH_4$ emissions with two distinct methane analysers (Los Gatos Research (LGR) UGGA and GASMET DX4040). The two analysers gave contradicting results, with apparent $CH_4$ fluxes differing both in direction and in magnitude (Fig 1). We hypothesised that these divergent measurements resulted from interferences of VOCs with $CH_4$ measurements. To test this hypothesis, we built a setup to quantify the effect of eleven different VOCs on five commonly used $CH_4$ analysers under controlled conditions. In this communication, we present results from field measurements and laboratory tests, as well as a first sensitivity analysis for the impact of VOC interferences on measurements of $CH_4$ fluxes from different ecosystem compartments.

## 2 Methods

### 2.1 Field measurements

Field measurements were conducted as part of a larger field campaign in the Skogaryd research forest in southern Sweden (58°23'N, 12°09'E) (Klemedtsson et al., 2010) in the summer of 2018. We measured spruce stem $CH_4$ emissions from 30

trees at different distances from the main ditch to achieve a gradient of water table levels. The trees were equipped with box chambers to measure stem gas exchange as described in Machacova et al. (2016). $CH_4$ emissions were measured by closing chambers for 20 minutes and recycling air through one of two portable analysers, a Los Gatos Research (LGR) UGGA OA-ICOS based $CH_4/CO_2/H_2O$ analyser and a Gasmet DX4040 FTIR based multi-compound analyser. $CH_4$ exchange rates were quantified as the increase in $CH_4$ mixing ratio over time, divided by the chamber volume and the stem area. Negative fluxes indicate a net $CH_4$ uptake and positive fluxes a net $CH_4$ release to the atmosphere. Measurements were conducted daily from June $2^{nd}$ to $13^{th}$ and from July $25^{th}$ to August $5^{th}$ 2018, alternating between the two instruments. In addition, we measured soil $CH_4$ fluxes from 9 soil collars (0.26 m$^2$) using a static chamber technique described previously (Klemedtsson et al., 2010). Measurements were conducted daily between June $2^{nd}$ and $13^{th}$, again alternating between the LGR UGGA and Gasmet DX4040 analysers.

## 2.2 Laboratory tests 1 – Qualitative screening for VOC interferences

In a first series of experiments, we qualitatively screened for VOCs that interfered with $CH_4$ analysers. We constructed an experimental system where VOCs can be added to an air stream with a constant $CH_4$ mixing ratio (Fig. 2a). Air from the in-house pressured air supply (compressed outdoor air) was first passed through a membrane drier (SMC IDX-series) and a zero-air generator (HPZA 3500 220, Parker Balston) to remove any VOCs present in the background air. Due to a defect, the zero-air generator did not remove $CH_4$ from the air source, such that the air used for our experiments contained atmospheric $CH_4$ at atmospheric mixing ratios.The air was then passed through a needle valve and a flow meter to set and monitor its flow rate. Next, we used two electronic three-way solenoid valves (SMC VX3-series) operated through a python script to guide the air flow either through a VOC source or a bypass line. The VOC source was an open or partly open vial that contained a pure VOC standard placed in a 500 mL glass bottle. The air flow was alternatingly set to the VOC source and bypass for 2.5 minutes. Finally, the air flow was passed to six instruments and an overflow outlet through T-connectors. All wetted parts of the air line after the zero-air generator were either stainless steel, PTFE or glass to prevent generation or removal of VOCs in the air flow path.

The flow rate of air entering the system was set slightly above the total air intake of all analysers (approximately 5 L min$^{"-1"}$). We tested four analysers based on laser spectroscopy (CRDS), including two stationary instruments (Picarro G2301 ($CO_2$, $CH_4$, $H_2O$); Picarro G2201i ($^{13}CO_2$, $^{13}CH_4$, $H_2O$) and two portable instruments (Picarro G4301; LGR UGGA ($CO_2$, $CH_4$, $H_2O$)), as well as a Fourier-transformed infrared (FTIR) spectroscopy based multi-compound analyser (GAS-MET DX4015). For control, we quantified VOC concentrations with a proton transfer reaction quadrupole mass spectrometer (PTR-MS, Ionicon Analytik GmbH). We used the system to test the interferences of 8 VOCs ($\alpha$- and $\beta$-pinene, $\Delta$3-carene, limonene, linalool, trans-2-hexenylacetate, cis-3-hexen-1-ol, nonanol, toluene, and methanol). Additional experiments with $\beta$-caryophyllene and nonanol were unsuccessful because the volatility of these compounds was too low, i.e., the mixing ratios generated for these compounds remained <50 ppbv. We chose the tested VOCs to represent a cross-section of naturally occurring VOCs and aimed to cover a wide range of chemical compound classes rather than the most important biogenic VOCs occurring in any given environment.

The Gasmet DX4015 analyser was used in the same way it was deployed for soil flux measurements in previous studies: spectra were measured over 5 seconds and deconvoluted based on a library with 4 compounds ($CH_4$, $H_2O$, $CO_2$, $N_2O$). Measurements at all instruments were averaged over 10 sec intervals.

## 2.3   Laboratory tests 2 – Quantification of VOC interferences

In a second series of experiments, we aimed to quantitatively measure VOC interferences. We modified the experimental setup such that VOC mixing ratios of the air passed to the $CH_4$ analysers could be controlled (Fig. 2b). VOC-free air and VOC carrying air were regulated separately by two mass flow controllers (Bürkert GmbH) and mixed through a T-connector. The flow rate of VOC free air was kept constant at 1 L $min^{-1}$ while the flow rate of the VOC carrying air was varied between 0 and 50 mL $min^{"-1"}$. The resulting flow rate, however, was too low to operate more than two instruments in parallel. We therefore

alternated between three $CH_4$ analysers (Picarro G2301, LGR UGGA, GASMET DX 4040) while continuously monitoring the VOC mixing ratios with the PTR-MS. For this second series of experiments, we replaced the FTIR-based analyser with a portable but otherwise similar model (GASMET DX4040) and increased the measurement cycle to one minute. The analyser was zero-calibrated with $N_2$ gas daily.

The PTR-MS was calibrated with a gas standard containing methanol, toluene, $\alpha$-pinene (presenting also other monoter-

penes: $\beta$-pinene, carene and limonene), cis-3-hexenol/hexanal as well as other VOCs not measured in this study. The mixing ratios of the other measured compounds were calculated based on the transmission curve obtained from the calibration (Taipale et al., 2008). Instruments were challenged with both gradual increases (Fig. 4) and step-wise changes (Fig. 5) of VOC mixing ratios, with 2-3 repetitions per instrument and test type. We tested six VOCs: $\beta$-pinene, $\Delta$3-carene, linalool, trans-2-hexenylacetate, cis-3-hexen-1-ol, and methanol.

## 2.4   Data analysis

FTIR spectra were deconvoluted using the software Calcmet to quantify the concentrations of methane and other trace gases. During Experiment 1, only $CO_2$, $H_2O$, $CH_4$ and $N_2O$ were included in the spectra library (i.e., interfering VOCs were not included in the spectral library). We acknowledge that this is not a correct application of the analyser in the presence of known interference according to the manufacturers guidelines. We did so to evaluate the impact of VOCs missing in the spectral library due to unexpectedly occurring VOCs, unidentified compounds, or user errors on $CH_4$ flux measurements.

During experiment 2 and for the field measurements, we separately quantified the effect of adding a VOC present or missing in the spectral library. To do so, we analyzed the data twice, once with limited library ($CO_2$, CO, $N_2O$, $H_2O$, $NH_3$) that did not contain the interfering VOCs, and once with a full library that contained spectra of all tested VOCs (additional compounds: methanol, a-pinene, b-pinene, carene, linalool, hexenol, nonanal, trans-2-hexenyl acetate, caryophyllene, limonene).

Interferences were calculated as the slope between VOC mixing ratio and apparent $CH_4$ mixing ratio. To avoid effects of transient interferences, we excluded time points where VOC mixing ratios abruptly changed (>35% change in VOC mixing ratio per minute). Repeated challenges with the same test were combined in one regression analysis, but step-wise and gradual challenges were analysed separately. We calculated conservative estimates of uncertainty taking into consideration the un-

certainty of the regression slope which already incorporates the variance among replicate tests. Our estimate of uncertainty furthermore accounts for minor variation in the $CH_4$ concentrations in the in-house pressurised air supply, which limited our ability to detect small interferences. We used a bootstrap approach to calculate this uncertainty. For this, the measured $CH_4$ concentrations were replaced by those from a random period of the same length during when no experiments were conducted (i.e., air contained no VOC at this time and all observed variations in $CH_4$ concentrations represented true changes in $CH_4$ concentrations). This approach was repeated a total of 500 times. The $50^{th}$, $97.5^{th}$, and $2.5^{th}$ percentiles of the slope between these simulations was subtracted from the upper and lower limit of the confidence interval found in the regression analysis to obtain the central 95% confidence interval for the interference. Significant interference was assumed when these confidence intervals did not include zero.

FTIR measurements with libraries that included the tested VOCs also reported concentration for these VOCs. To evaluated the viability of measuring VOC concentrations by FTIR, we calculated the regression between VOC concentrations measured by FTIR and PTR-MS. We note that we made no attempts to calibrate FTIR based VOC concentration against external standards. All statistical analysis was conducted in the statistical programming environment R version 3.4.4 (R Development Core Team, 2015). All stated uncertainties refer to 95% confidence intervals.

## 2.5 Impact assessment for soil, stem, and shoot chambers

We assessed of the potential impact of VOC interferences on $CH_4$ flux measurements in three scenarios representing soil, stem, and shoot chamber measurements. The assumptions used for these estimates are shown in Table 1. Chamber dimensions and $CH_4$ and VOC flux rate, were chosen based on measurements conducted at SMEAR II LTER field station (Hyytiälä, Finland) (Hari and Kulmala, 2005).

Only monoterpens (PTR/MS signal at m/z 137) were taken into account, and it was assumed that these VOCs uniformly interfered with $CH_4$ measurements at the same rate as $\beta$-pinene. We furthermore assumed that VOC emission rates remain constant over the chamber closure time, i.e., that chamber headspace VOC mixing ratios do not approach saturation during the closure. While this assumption is unlikely to hold true for shoot chambers, it allows us to conduct a worst case estimate for VOC interferences. For each chamber type, we assessed the effects of VOC emissions at typical (i.e., average) as well as peak (maximum) emission rates. For FTIR, were estimated the effects of both VOCs present in the spectral library (interference measured on DX4040 with full library) and VOCs missing in the spectral library (interference on DX4040 with limited library).

Based on these assumptions, we calculated the actual change in $CH_4$ mixing ratios during a chamber closure, the VOC mixing ratio reached at the end of the chamber closure, the upper limit to the apparent $CH_4$ mixing ratio measured due to VOC interference on each analyser, and the maximum ratio of apparent to actual $CH_4$ emissions. We emphasise that this is only a preliminary assessment of the impact of VOC interferences on $CH_4$ flux measurements, as neither the identity of all emitted VOCs nor their interference on different analysers are fully known. These results of these calculations should therefore be understood as order-of-magnitude estimates.

## 3   Results

### 3.1   Initial analysis of field data

Our initial spruce stem measurements showed a stark discrepancy between stem $CH_4$ emissions measured with the LGR UGGA and the GASMET DX4040 analysers. Measurements conducted with the LGR UGGA ranged from an apparent $CH_4$
uptake of -2 $\mu$g $CH_4$ h$^{-1}$ m$^{-2}$ and an apparent $CH_4$ emission of 7 $\mu$g $CH_4$ h$^{-1}$ m$^{-2}$ (Fig 1). Measurements conducted with the DX4040 (limited spectral library) consistently showed an apparent $CH_4$ uptake ranging with a much larger flux (-145 to +8 $\mu$g $CH_4$ h$^{-1}$ m$^{-2}$). The average $CH_4$ fluxes were +0.44 $\pm$ 0.15 $\mu$g $CH_4$ h$^{-1}$ m$^{-2}$ (LGR UGGA) and -17.4 $\pm$ 3.7 $\mu$g $CH_4$ h$^{-1}$ m$^{-2}$ (GASMET DX4040). In contrast, both analysers measured similar soil $CH_4$ fluxes, with average fluxes of -36.0 $\pm$ 7.9 (LGR UGGA) and -19.4 $\pm$ 5.3 $\mu$g $CH_4$ h$^{-1}$ m$^{-2}$ (GASMET DX4040).

### 3.2   Qualitative screening for interferences

An example for the changes in VOC mixing ratios over time produced by our setup is shown in Fig. 3a. The installation was first operated without a VOC present in the source to control for artefacts (e.g., effects of pressure changes due to switching valves). At the time point indicated by the vertical dashed line, a vial with $\beta$-pinene was inserted into the VOC source. This resulted in periodic patterns of presence and absence of $\beta$-pinene in the analysed air stream, with a maximum mixing ratio of approximately 5 ppmv.

The response of the $CH_4$ analysers to the changing $\beta$-pinene mixing ratios is depicted in Fig. 3b-h. The FTIR-based analyser (DX4040) showed the strongest interference, with $CH_4$ readings reaching by up to 4 ppmv when $\beta$-pinene was added to the air stream, i.e., 2 ppmv above the actually $CH_4$ mixing ratio (Fig. 3b). In contrast, measured $CH_4$ mixing ratios remained stable around 2ppmv when setup was operated with an empty vial in the VOC source, demonstrating that the observed interferences were not artefacts produced by the experimental setup (i.e., pressure effects).

The Picarro G2301 analyser exhibited moderate interferences by *changes* in VOC mixing ratios (Fig. 3c). The sudden increase in the $\beta$-pinene mixing ratios resulted in temporary positive deviations corresponding to 20 ppbv $CH_4$ ppmv$^{-1}$ $\beta$-pinene. We also detected a negative deviation when VOCs were suddenly removed from the air stream. A similar, but much weaker (~1ppbv) interference was also detected on the Picarro G2201i instrument (Fig. 3d). The LGR UGGA and the Picarro G4301 instruments showed no discernible effect of the addition of $\beta$-pinene to the air stream (Fig. 3e-f), however, for the G4301 analyser this was because relatively high noise and occasional outliers in the measured $CH_4$ mixing ratio may have masked potential small interferences. Finally, we did not detect any interference of $\beta$-pinene with the measured $\delta^{13}C_{CH4}$ values (Fig. 3g).

An overview of the interference tests with other VOCs is provided in Table 2. Among the 11 tested compounds, 9 showed an interference with the DX4015 analyser, 8 with the Picarro G2301, 6 with the Picarro G2201i, and 3 with the LGR UGGA. Interferences on the DX4015 were typically 2 orders of magnitude higher than on laser absorption based analysers. All interferences with $CH_4$ mixing ratio measurements on the Picarro G2301 and G2201i instruments were transient, similar to those shown for $\beta$-pinene (Fig. 3c).

Only two VOCs interfered with $\delta^{13}C_{CH4}$ measurements by the Picarro G2201. First, toluene, which was added at high mixing ratios (30 000 - 35 000 ppmv) lead to an apparent increase in $\delta^{13}C_{CH4}$ values by 1‰. Second, an accidental addition of high mixing ratios of methanol (>80 000 ppbv, likely higher due to saturation of the PTR-MS) strongly interfered with $\delta^{13}C_{CH4}$ measurements, leading to a positive deviation by about 900‰ with a memory effect that lasted more than 2 hours
(not shown).

## 3.3   Quantification of interferences

In our second experiment, we successfully created gradual and step-wise changes in VOC mixing ratios. As an example, the effects of gradual and step-wise changes in $\beta$-pinene mixing ratios on the apparent $CH_4$ mixing ratios measured by three different analysers are shown in Fig. 4a and Fig. 5a, respectively. In this experiment, we did not detect a significant effect of
$\beta$-pinene mixing ratios on $CH_4$ mixing ratios measured with the Picarro G2301 (Figs. 4b,5b) or the LGR UGGA instruments (Figs. 4e,5e). In contrast, $\beta$-pinene led to a significant underestimation of $CH_4$ mixing ratios with the Gasmet DX4040 (by approximately 120 ppbv $CH_4$ ppmv$^{-1}$ $\beta$-pinene) when $\beta$-pinene was not part of the spectral library (Figs. 4lc,5c). Including $\beta$-pinene (and other VOCs) in the spectra library significantly reduced this interference to approximately 1 ppbv $CH_4$ ppmv$^{-1}$ $\beta$-pinene (Figs. 4d,5d).

Similar results were found in tests with other VOCs. A list of the interferences quantified in different experiments is provided in Table 3. We did not detect a significant effect of VOC mixing ratios on the apparent $CH_4$ mixing ratios measured by the Picarro G2301 and the LGR UGGA. For $\beta$-pinene and $\Delta$3-carene we constrained the upper confidence limits were <1 ppbv $CH_4$ ppmv$^{-1}$ VOC on both instruments, for other compounds confidence limits were higher, mainly due to lower mixing ratios during the tests.

Interference on the Gasmet DX4040 without specific libraries for the tested compounds were high, ranging from -35 ppbv ppmv$^{-1}$ (methanol) to 1800 ppbv ppm$^{-1}$ (cis-3-hexen-1-ol). Adding reference spectra of the tested VOCs to the library substantially decreased the interferences, but significant interferences were still detected for $\beta$-pinene, 3-carene and hexenylacetate. (Table 3).

FTIR- and PTR-MS based measurements of VOC mixing ratios were highly correlated (R=0.956 to 0.998) for most com-
pounds (Fig. 6). Poor correlations were found for linalool, which was present at mixing ratios close to or below the detection limit of the FTIR method (10 ppbv).

## 3.4   Revised analysis of field data

After re-analysis with the full library, our field measurements by FTIR showed smaller $CH_4$ fluxes than in our initial analysis (Fig. 1). The methane emission rates generated in this revised analysis (-85 to +8 $\mu$g $CH_4$ h$^{-1}$ m$^{-2}$), however, still showed a
substantial net uptake of $CH_4$. The average apparent $CH_4$ flux was -10.1 $\pm$ 1.6 $\mu$g $CH_4$ h$^{-1}$ m$^{-2}$. Assuming that measurements conducted by OA-ICOS revealed the true $CH_4$ flux, the re-analysis decreased the bias in FTIR based measurements by 41%. In contrast, the re-analysed of soil $CH_4$ fluxes resulted in slightly lower average flux (-19.1 $\pm$ 6.1 $\mu$g $CH_4$ h$^{-1}$ m$^{-2}$) compared to initial measurements with the limited library (-19.4 $\pm$ 5.3 $\mu$g $CH_4$ h$^{-1}$ m$^{-2}$).

### 3.5 Estimated impact on static chamber systems on different ecosystem compartments.

VOC (monoterpene) to methane emission ratios increased from soil to stem to shoot chambers, spanning over four orders if magnitude (Table 1). The practical impact of VOC interferences on $CH_4$ strongly differed between ecosystem compartments. True $CH_4$ fluxes typically exceeded apparent $CH_4$ fluxes due to VOC interferences by 2 or more orders of magnitude in soil chambers, whereas the the upper limit of apparent $CH_4$ fluxes was equal or greater than true fluxes in shoot chambers (Fig. 7, Table 4).

Our impact estimates suggest the all analysers were able to accurately (<5% measurement error) quantify soil $CH_4$ fluxes at average VOC emission rates, even if important VOCs are missing in the FTIR spectral library (Fig. 7. Stem flux measurements, in contrast, are more vulnerable to VOC interferences, with upper limits of confidence on the order of 2-6% of the actual $CH_4$ flux, except for FTIR with incomplete spectral libraries where apparent $CH_4$ fluxes were estimated to exceed to interference may exceed actual fluxes several fold.

VOC interferences are a serious challenge for quantifying $CH_4$ flux in shoot chambers where VOC fluxes are approximately 4 orders of magnitude higher than $CH_4$ fluxes. Our results show that apparent fluxes due to VOC interferences can exceed actual fluxes when shoot $CH_4$ fluxes are measured by FTIR, even if all VOCs are included in the spectral library. While we were not able to detect significant VOC interferences on OA-ICOS and CRDS based analysers, the upper limit of uncertainty of these interferences still allows for interferences that exceed actual $CH_4$ fluxes in shoot chambers.

## 4 Discussion

### 4.1 FTIR-based analysers

Our results show that FTIR based analysers are not well suited for measuring plant $CH_4$ fluxes and other applications that quantify small changes in $CH_4$ mixing ratios in the presence of much larger changes in the mixing ratios of other compounds, as is the case for plant $CH_4$ flux measurements (Tab. 4, Fig. 7). In particular, our work emphasises that FTIR based $CH_4$ flux measurements can only provide reliable data if all VOCs that co-emitted in relevant amounts are identified and included in the spectral library.

Measurements of plant $CH_4$ emissions with incomplete spectral libraries can result in gross over- or under-estimations of the actual $CH_4$ flux rates depending on the combination of co-emitted VOCs as well as the components included in the spectral library used to deconvolute the measured spectra. The presence of VOCs missing in the spectral library is typically indicated by high residual values for the spectral fitting, such measurements should be re-analysed with an amended spectral library or, if this is not possible, considered invalid. Spectral libraries compiled for soil flux measurements are not sufficient for quantifying $CH_4$ fluxes from tree stems. Had we solely relied on an FTIR system with an incomplete spectral library intended from soil flux measurements to quantify $CH_4$ fluxes during our field campaign in Skogaryd, we would have identified spruce stems as a strong sink of $CH_4$ (Fig. 1). However, concurrent measurements by the OA-ICOS-based LGR UGGA, which were largely unaffected by VOC co-emissions (Table 3), revealed that these trees stems actually act as a small source of $CH_4$. The comparison of

OA-ICOS- and FTIR- based results indicates that tree stem VOC emissions at Skogaryd were dominated by compounds that negatively interfere with FTIR measurements $CH_4$ measurements, including methanol, $\beta$-pinene, and hexenylacetate. The effect of these VOCs outweighted the positive interference of other VOCs including $\Delta$3-carene and hexenol. It is, however, important to note that we did not quantify the interfereces of all potential VOCs, including the dominant compound emitted by spruce trees ($\alpha$-pinene) (Grabmer et al., 2006; Janson, 1993).

Our second experiment further showed that the VOC interferences can be minimized by including all potentially occurring VOCs in the spectral library. In our experiments, this decreased the interference by 1-2 orders of magnitude 3. This, however, may not be practical in many field settings, where the identity of VOCs released from plants and soils is often unknown. Furthermore, spectral deconvolution was not successful for all VOCs, and significant interferences were found for three of the tested VOCs ($\beta$-pinene, $\Delta$3-Carene, and hexenyl acetate) even when the reference spectra were present in the spectral library. Upper limits for the quantified interferences in FTIR-based measurements were typically an order of magnitude higher than on laser absorption based instruments. In the case of our field campaign in Skogaryd, on average 59% of the interference persisted when data were re-analysis with additional spectra in the library (Fig. 1).

In contrast, FTIR and OA-ICOS based analysers measured similar $CH_4$ fluxes from soil chambers. This shows that both measurement principles can reliably quantify soil $CH_4$ fluxes, where the VOC:methane flux ratio is significantly lower than in tree stems and shoots, which is consistent with previous studies (e.g. Falk et al., 2014). Our study furthermore showed that FTIR-based analysis may be a useful method to study VOC fluxes instead of or in addition to PTR-MS measurements. The strong correlation between VOC mixing ratios quantified by FTIR and PTR-MS (Fig 6) indicates that FTIR can conduct precise measurements of VOC mixing ratios. FTIR instruments are cheaper and more portable than PTR-MS instruments and provide a complementary analytical principle that could help distinguish between isomers that cannot be separated by mass spectrometry. Detection limits of FTIR based measurements of VOC mixing ratios (10s of ppb), however, are substantially higher than those of PTR-MS based measurements (10s of ppt), and cross sensitivities among VOCs may bias the quantification of compounds that occur at lower mixing ratios.

## 4.2 Laser spectroscopy based analysers

Interferences on the CRDS- and OA-ICOS- based systems were significantly lower than on FTIR-based systems, but during our qualitative screening we still detected some potentially important interferences (Fig. 3), especially the case for the Picarro G2301. On this analyser, sudden changes in the VOC mixing ratio resulted in minor deviations of the measured $CH_4$ mixing ratios. These interferences, however, were corrected by the instrument over the course of approximately 30 sec and are therefore unlikely to affect chamber measurements, where mixing ratios of VOCs and $CH_4$ increase gradually (e.g., over a 20–40 minutes chamber closure). These interferences may, however, pose an important bias for measurements that rely on fast measurements of air masses with changing VOC mixing ratios as used for eddy covariance (EC) measurements. In these measurements, interferences from VOC emissions as detected in this study could potentially lead to an overestimation of $CH_4$ emissions. We have, however, not been able to further investigate VOC interferences on the high-frequency analysers used for EC measurements.

## 5 Conclusions

We quantified the interference of VOCs on $CH_4$ analysers based on FTIR and laser absorption spectroscopy. FTIR based instruments were more prone to higher levels of interference than laser absorption based instruments, even when VOCs were added to the spectral library. FTIR based analysers are therefore not well suited for studies of plant $CH_4$ fluxes and other applications where small $CH_4$ fluxes need to be quantified in the presence of much higher fluxes of VOCs. Our results, however, also indicate that FTIR instruments can be a cost-effective solution to field measurements of certain VOCs.

*Code and data availability.* Raw data, processed data, and code are available at doi:10.5281/zenodo.2597716.

*Author contributions.* LK had the main responsibility for analysing the data and writing the manuscript; and participated in the designing and construction of the measurement setup. MK had the main responsibility for designing the measurement setup and programming the controlling software; and participated in constructing the measurement setup and in the writing process. KR had the main responsibility for the VOC measurements and processing of PTR-MS results. IH had the main responsibility for the field campaign and had the original idea for testing the interference of VOCs in $CH_4$ analysers; and participated in designing the measurement setup. TP had the main responsibility for constructing the measurement setup; and participated in designing the measurement setup. HH contributed to the conceptualisation of the study and was responsible for deciding and providing the measured VOCs. MP contributed to the conceptualisation of the study and the writing of the manuscript.

*Competing interests.* The authors declare no conflict of interest.

*Acknowledgements.* This project has received funding from the European Research Council (ERC) under the European Union's Horizon 2020 research and innovation programme (grant agreement number 757695) and the Academy of Finland (grant numbers 319329 and 2884941). We thanks Gasmet Technologies Oy and Annalea Lohila for providing access to FTIR analysers.

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

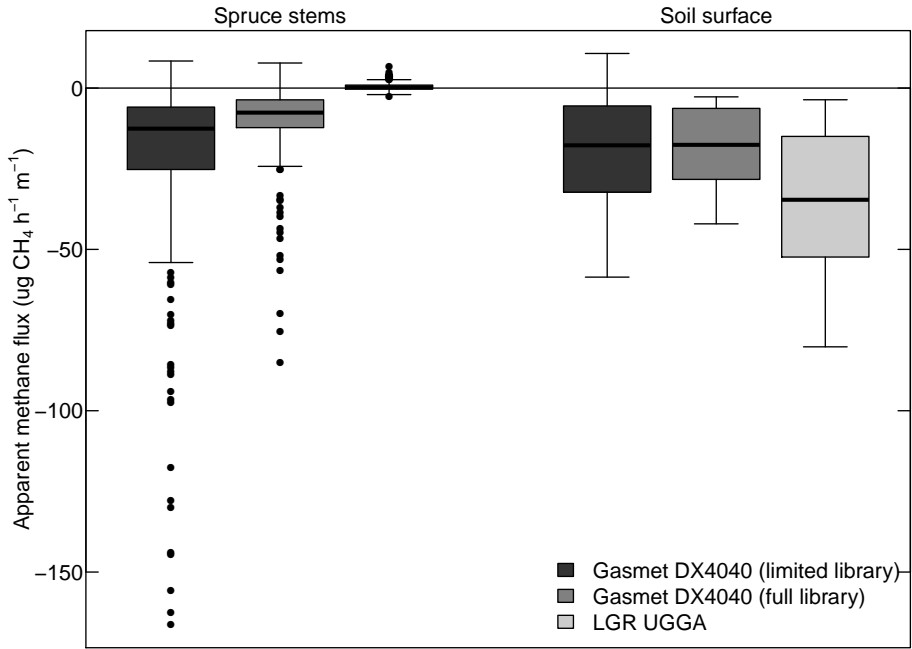

**Figure 1.** Apparent tree stem methane fluxes when quantified with a laser spectroscopy based analyser (LGR UGGA) and a FTIR based analyser (Gasmet DX4040). FTIR based fluxes are shown calculated based on spectral deconvolution with a minimal library that did not contain VOC spectra (min. lib.), and with a library that contained spectra of commonly occurring VOCs (full lib.).

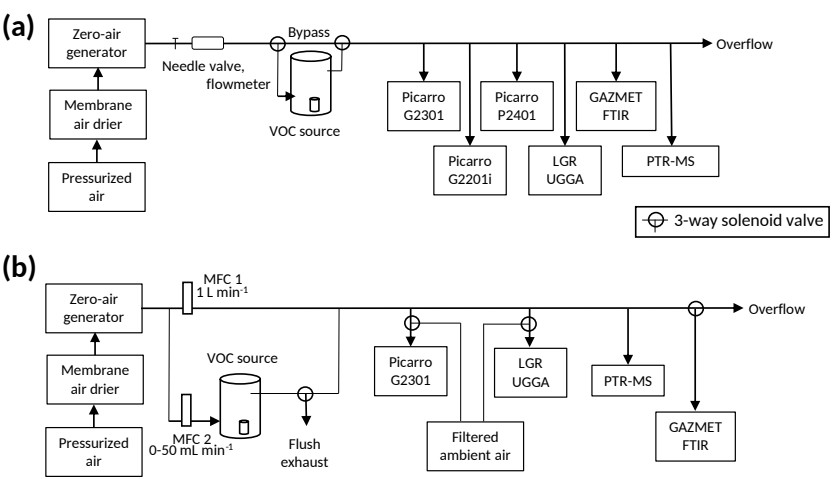

**Figure 2.** Schematic for air flow in laboratory test 1 (panel **a**) and 2 (panel **b**).

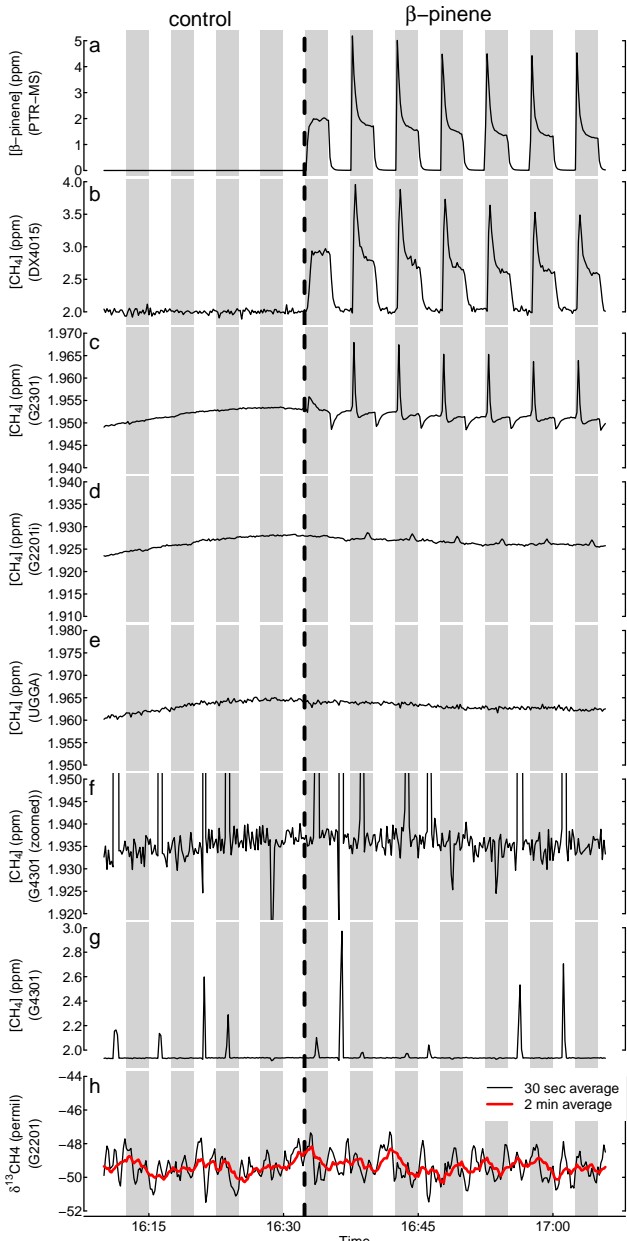

**Figure 3.** Exemplary results from Experiment 1, shown for tests conducted with $\beta$-pinene. The panels show the development of the $\beta$-pinene (panel **a**) mixing ratio as measured by PTR-MS and apparent $CH_4$ mixing ratio as measured by Gasmet DX4015 (using an incomplete library intended for soil flux measurements), Picarro G2301, Picarro G2201i, LGR UGGA and Picarro G4301 (panels **b–g**, respectively) and $\delta^{13}C$-$CH_4$ values as measured by Picarro G2201i (panel **h**). White areas indicate the times when the system was set to bypass the VOC source, grey shaded areas times when the VOC source was online. During the control period left of the dashed vertical line the VOC source was empty. At the position of the dashed vertical line, $\beta$-pinene vial was introduced into the standard source. Black line represents 10-second moving average of apparent $CH_4$ mixing ratios and $\delta^{13}C_{CH4}$ values, red thick line 30-second moving average of appearent $\delta^{13}C_{CH4}$ values. Notice G4401 results zoomed in panel **f** to visualise background variation; full-scale results in panel **g**.

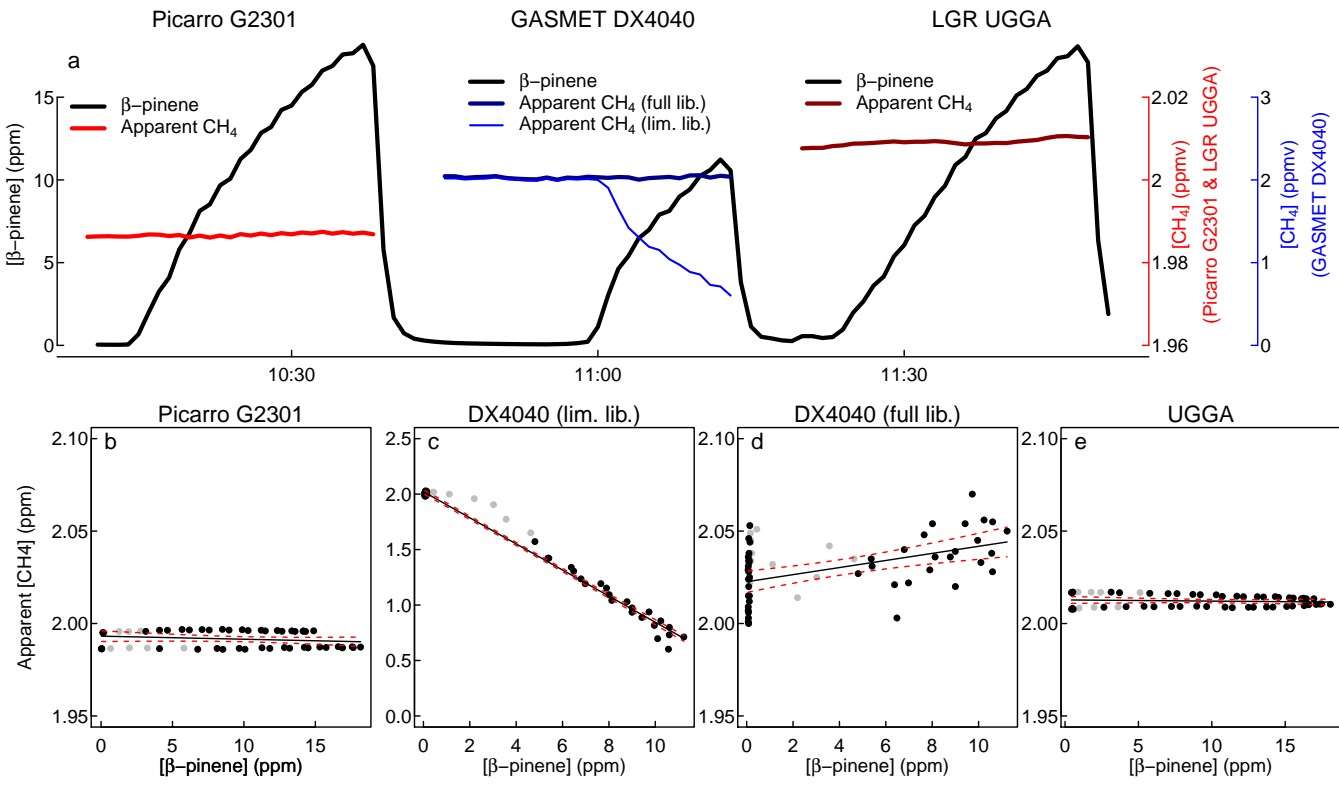

**Figure 4.** Quantitative measurements of the effect of $\beta$-pinene mixing ratios on measured (apparent) methane mixing ratios when analysers were challenged with a gradual increase in the $\beta$-pinene mixing ratio . The figure depicts an example for the time course of $\beta$-pinene and apparent $CH_4$ mixing ratios (**a**) as well as the relationship between $\beta$-pinene and the measured $CH_4$ mixing ratio (**b-e**). Note that in panel a, $CH_4$ concentrations measured by the Gasmet DX4040 analyser are depicted on a different scale (blue) than those measured by the Picarro G2301 and LGR UGGA analysers (red). Black lines in panels b-e indicate linear regressions, dashed red lines the 95% confidence interval of these regressions. Data points that occurred after after a rapid changes in the $\beta$-pinene mixing ratio and that were therefore excluded from the regression analysis are depicted in grey.

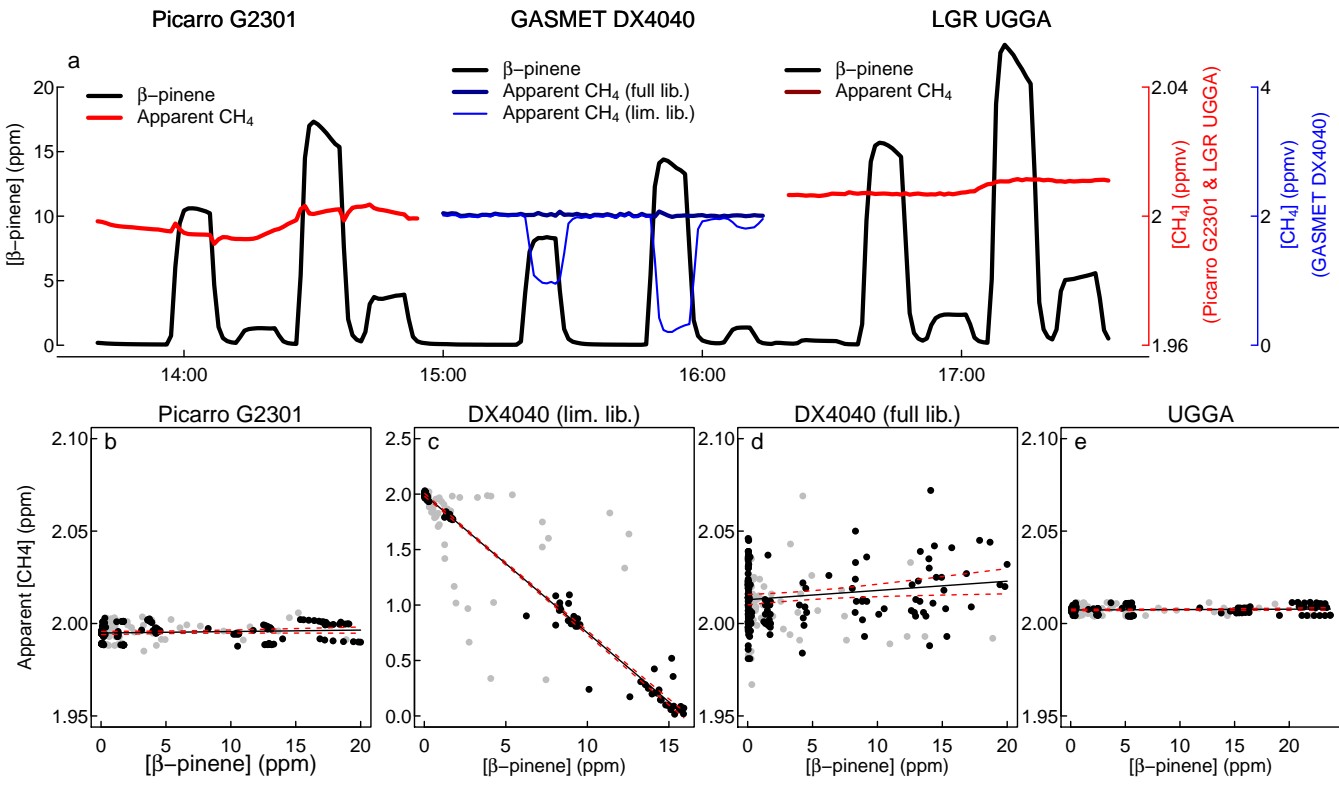

**Figure 5.** Quantitative measurements of the effect of $\beta$-pinene mixing ratios on measured (apparent) methane mixing ratios when analysers were challenged with stepwise changes in the $\beta$-pinene mixing ratio . The figure depicts an example for the time course of $\beta$-pinene and apparent $CH_4$ mixing ratios (**a**) as well as the relationship between $\beta$-pinene and the measured $CH_4$ mixing ratio (**b-e**). Note that in panel a, $CH_4$ concentrations measured by the Gasmet DX4040 analyser are depicted on a different scale (blue) than those measured by the Picarro G2301 and LGR UGGA analysers (red). Black lines in panels b-e indicate linear regressions, dashed red lines the 95% confidence interval of these regressions. Data points that occurred after after a rapid changes in the $\beta$-pinene mixing ratio and that were therefore excluded from the regression analysis are depicted in grey.

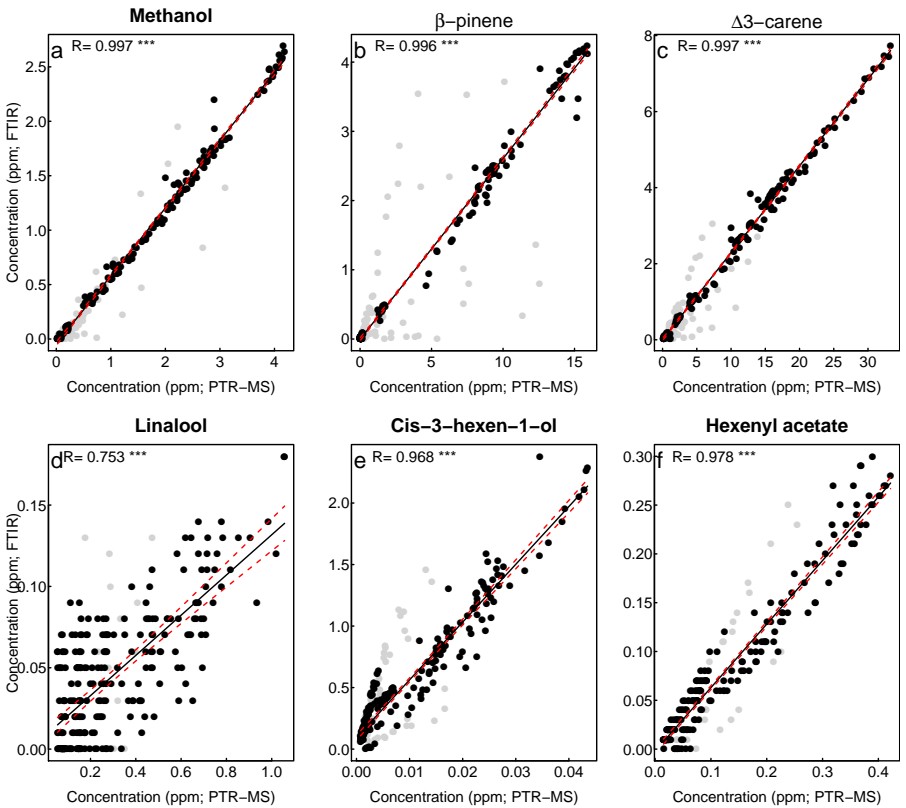

**Figure 6.** Correlation between FTIR- and PTR-MS based measurements of VOC mixing ratios. Data points plotted in grey were excluded after rapid changes in the VOC mixing ratio. Asterisks indicate significant levels: *, p<0.05; **, p<0.01; ***, p<0.001.

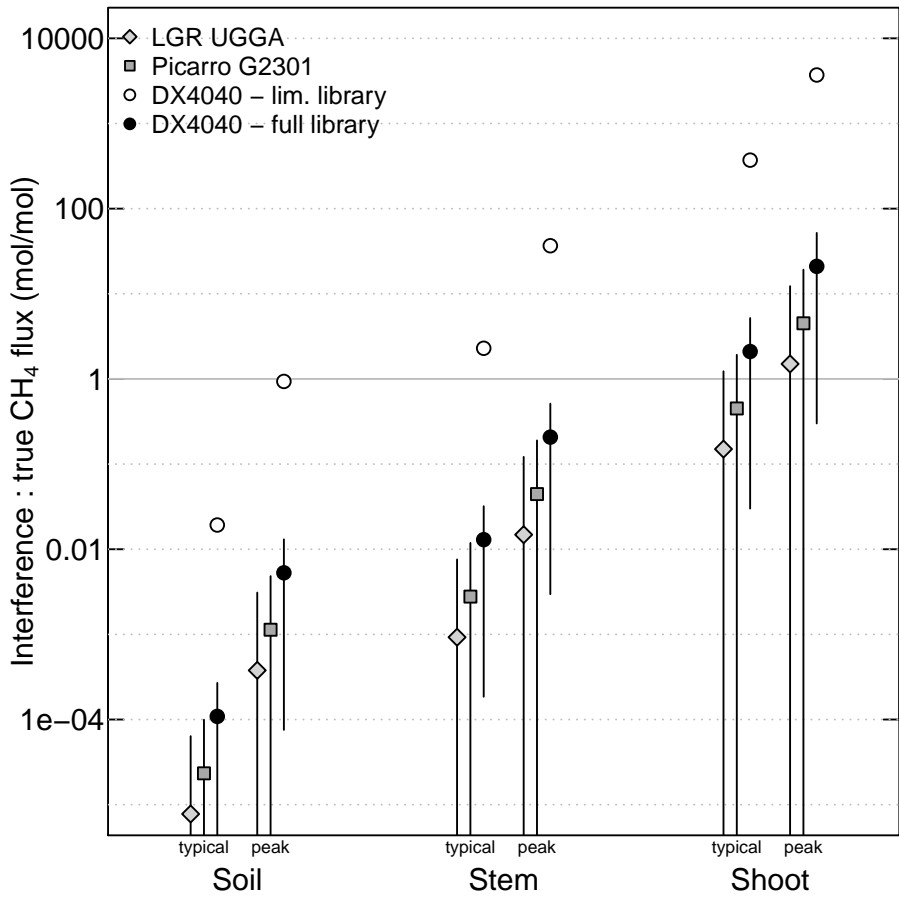

**Figure 7.** Estimated size of the $CH_4$ flux measurement error due to VOC interference (at typical and peak VOC fluxes) relative to the mean actual $CH_4$ fluxes in soil, stem, and shoot chambers. Assumptions underlying these estimates are shown in Table 1. Only monoterpens (m/z=137 in PTR-MS measurements) were taken into account for this estimate, and it was assumed that all monoterpens interfere with CH4 analysers the same rate as $\beta$-pinene. The results presented here should therefore be understood as order-of-magnitude estimates. Symbols indicate medians with error bars indicate the analytical uncertainty (95% confidence interval) associated with the quantification of VOC interferences (but do not take into account uncertainties in other assumptions).

**Table 1.** Assumptions used to estimate VOC effects on $CH_4$ flux measurements in static soil, stem, and shoot chambers. Where available, assumptions are based on measurements conducted in the Scots pine forest at the SMEAR II research station (Hyytiälä, Finland).

| Chamber type | Soil chamber (upland) | | Stem chamber | | Shoot chamber | |
|---|---|---|---|---|---|---|
| VOC emission scenario | typical | peak | typical | peak | typical | peak |
| Chamber volume (L) | 100 | | 1 | | 2 | |
| Soil/stem surface ($m^2$) or foliage biomass (g d.w.) per chamber | 0.3 | | 0.01 | | 10 | |
| Closure time (min) | 10 | | 10 | | 10 | |
| Mean $CH_4$ emission rate ($\mu$mol m$^{-2}$ h$^{-}$1 or $\mu$mol g$^{-1}$ d.w. h$^{-}$1) | -0.90[1] | | 0.027[2] | | 0.0005[3] | |
| Monoterpene emission rate ($\mu$mol m$^{-2}$ h$^{-}$1 or $\mu$mol g$^{-1}$ d.w. h$^{-}$1) | 0.14[4] | 6.8[4] | 0.5[5] | 8[5] | 1.5[6] | 15[6] |
| Monoterpene:$CH_4$ emission ratio (mol/mol) | -0.15 | -7.6 | 19 | 300 | 3 000 | 30 000 |

Sources: [1] Machacova et al. (2016) [2] Machacova et al. (2016) [3] Estimate based on Keppler et al. (2006)) [4] Aaltonen et al. (2013) [5] Vanhatalo et al. (2015); Rissanen et al. (2016) [6] Tarvainen et al. (2005)

**Table 2.** Summary of interferences detected in qualitative tests

| Compound | | Interference (ppbv apparent $CH_4$) | | | | |
|---|---|---|---|---|---|---|
| name | conc. range (ppbv) [ion] | Gasmet DX4015 | Picarro G2301 | Picarro G2201i | Picarro G4301 | LGR UGGA |
| Methanol | 6 000 - 10 000 [33] | 500 - 700 | $15^a$ | $2^a$ | – | 2 |
| $\alpha$-pinene | 4 000 - 5 000 [137] | 1 500 – 2 000 | $10\text{-}15^a$ | $1^a$ | – | – |
| $\beta$-pinene | 5 000 - 15 000 [137] | 2 000 | $5\text{-}30^a$ | $1^a$ | – | – |
| Carene | 3 000 - 7 000 [137] | 7 000 - 12 000 | – | – | – | |
| R(+)limonene | 900 - 1 100 [137] | 400 - 500 | $5^a$ | – | – | – |
| Linalool | 7 000 – 12 000 [155] | 300 - 600 | $8\text{-}25^a$ | $3\text{-}8^a$ | – | 0-8 |
| Cis-3-hexen-1-ol | 20-60 [101] | 600 – 3 000 | $10\text{-}15^a$ | – | – | – |
| Trans-2-hexenyl acetate | 500 – 2 000 [143] | 600 - 2 600 | $10\text{-}50^a$ | $2\text{-}12^a$ | – | – |
| Toluene | 30 000 – 35 000 [93] | 5 000 – 10 000 | $200\text{-}250^a$ | $15\text{-}20^a$ | – | 2 |

–, not detected

[a] Transient interference triggered by change in VOC mixing ratio rather that presence of VOC

**Table 3.** Quantified interferences of volatile organic compounds on $CH_4$ analysers. Significant interferences are indicated indicated in bold.

| | | Interference (ppbv apparent $CH_4$ per ppmv VOC; 95% CI) | | | |
|---|---|---|---|---|---|
| | | Picarro G2301 | LGR UGGA | Gasmet DX 4040 (full library) | Gasmet DX 4040 (lim. library) |
| Methanol | stepwise | 0.37 | 0.25 | 3.49 | **-35.8** |
| | | (-2.69 - 3.77) | (-3.25 - 3.33) | (-1.06 - 8.02) | **(-40.4 - -31.3)** |
| | gradual | 3.88 | 1.33 | 2.66 | **-36.6** |
| | | (-7.76 - 9.71) | (-5.91 - 6.36) | (-9.37 - 10.7) | **(-48.6 - -28.6)** |
| $\beta$-pinene | stepwise | 0.15 | 0.05 | **0.70** | **-123.8** |
| | | (-0.28 - 0.64) | (-0.29 - 0.41) | **(0.01 - 1.73)** | **(-125.5 - -122.0)** |
| | gradual | -0.12 | -0.06 | 1.94 | **-118** |
| | | (-1.82 - 0.74) | (-1.28 - 0.82) | (-0.12 - 3.41) | **(-122 - -114)** |
| $\Delta$3-Carene | stepwise | 0.22 | 0.10 | **4.23** | **64.8** |
| | | (-0.65 - 0.77) | (-0.64 - 0.78) | **(3.15 - 5.13)** | **(63.4 - 65.9)** |
| | gradual | -0.18 | -0.16 | **3.40** | **63.2** |
| | | (-1.28 - 0.53) | (-1.27 - 0.51) | **(2.04 - 4.34)** | **(61.3 – 64.6)** |
| Linalool | stepwise | 2.26 | -1.12 | 17.4 | -12.0 |
| | | (-15.1 - 18.0) | (-16.1 - 13.7) | (-7.80 - 40.3) | (-36.1 - 9.88) |
| | gradual | 19.8 | -0.16 | 17.7 | -14.8 |
| | | (-17.8 - 79.4) | (-33.2 - 20.7) | (-26.0 - 65.9) | (-58.3 - 33.6) |
| Cis-3-hexe-1-nol | stepwise | 4.80 | -5.81 | 477 | **1800** |
| | | (-431 - 229) | (-387 - 275) | (-105 - 903) | **(1230 – 2210)** |
| | gradual | 36.3 | 15.6 | 646 | **2210** |
| | | (-692 - 277) | (-802 - 516) | (-350 - 1240) | **(1210 - 2810)** |
| Trans-2-hexenyl acetate | stepwise | 1.39 | 1.94 | **-42.6** | **-402** |
| | | (-15.1 - 21.3) | (-17.8 - 22.6) | **(-74.9 - -8.16)** | **(-439 - -362.4)** |
| | gradual | 1.95 | 2.83 | **-126** | **-742** |
| | | (-25.5 - 37.3) | (-40.8 - 34.2) | **(-190 - -63.8)** | **(-820 - -667)** |

**Table 4.** Estimated impact of VOC interferences on methane flux measurements based on literature data of $CH_4$ and VOC fluxes.

| Chamber type | | Soil chamber (upland) | | Stem chamber | | Shoot chamber | |
|---|---|---|---|---|---|---|---|
| VOC emission scenario | | typical | peak | typical | peak | typical | peak |
| $\Delta_{Monoterpene}{}^{a}$ (ppbv) | | 1.7 | 82 | 20 | 320 | 30 000 | 300 000 |
| Actual $\Delta_{CH_4}{}^{b}$ (ppbv) | | 11 | | 11 | | 11 | |
| Max. interference[c] | | | | | | | |
| ( ppbv $CH_4$) | Picarro G2301 | 0.0031 | 0.15 | 0.037 | 0.59 | 55 | 550 |
| | LGR UGGA | 0.0021 | 0.11 | 0.026 | 0.41 | 39 | 390 |
| | DX4040 (lim. library) | 0.0058 | 0.28 | 0.069 | 1.1 | 100 | 1000 |
| | DX4040 (full library) | 0.21 | 10 | 2.5 | 40 | 3 700 | 37 000 |
| Max. interference : actual flux[d] | Picarro G2301 | 0.00028 | 0.014 | 0.034 | 0.54 | 5.5 | 55 |
| | LGR UGGA | 0.00020 | 0.0097 | 0.024 | 0.38 | 3.8 | 38 |
| | DX4040 (lim. library) | 0.00053 | 0.027 | 0.063 | 1.0 | 10 | 100 |
| | DX4040 (full library) | 0.19 | 0.92 | 2.3 | 36 | 370 | 3700 |

[a] Monoterpene mixing ratios at the end of a chamber closure, estimated based on the flux rates, chamber characteristics, and closure times stated in Table 1. We assumed that fluxes remained constant throughout the chamber closure period. Monoterpene saturation in the chamber headspace may decrease monoterpene emission rates during chamber closure.

[b] Change in $CH_4$ mixing ratio during chamber closure, estimated based on assumptions stated in Table , estimated based on the flux rates, chamber characteristics, and closure times stated in Table 1.

[c] Upper confidence interval for the false $\Delta CH_4$ detected due to monoterpene interference with $CH_4$ mixing ratio measurements.

[d] Ratio of the error in $CH_4$ flux measrement due to monoterpene interference to the actual $CH_4$ flux.