# Peer review of "Technical note: Interferences of volatile organic compounds (VOC) on methane concentration measurements"

_Biogeosciences, 2019_

## Referee Comment (RC1) · Anonymous Referee #1 · 26 Apr 2019

General comments

The paper by Kohl et al. describes cross sensitivities of several volatile organic compounds on methane measurements when using different optical analysers. I consider the results of the paper of major interest to all those monitoring methane fluxes in the field or laboratory from ecosystems and biological systems that are known to release VOC at substantial amounts. I found the manuscript to be well written and structured. The results are clearly presented and discussed in a straightforward manner, providing the scientific community with important information about how emissions of VOC released from the biosphere might interfere with measurements of methane when us-

ing state of the art optical measurement systems. I recommend publication of the manuscript as a Technical Note in Biogeosciences after minor revisions. I have only a few comments which I hope the authors might consider in their revised manuscript.

Specific comments

I would suggest using ppmv/ppbv/pptv (parts per million/billion/trillion by volume) throughout the whole manuscript instead of ppm/ppb/ppt.

Furthermore, the correct expression for ppmv would be mole fraction. However, I also understand if the authors would like to keep the more commonly used term "concentration".

Methods: As water vapour might substantially affect measurements of methane (both concentrations and stable carbon isotopes) when using optical analyzers I would suggest to add a few sentences how the authors have dealt with this issue during their investigations in the field and in the laboratory.

Discussion: Please add some information what are typical emission rates of some VOC released from vegetation/trees in the field and put them into relation with the amounts that have been applied in the laboratory study.

Figure 4: There are too many subfigures included and for some subfigures it is rather difficult to decipher the information. Please revise and split into two or three figures to increase readability.

Technical corrections

Page 5, line 6, Results: add CH4 after 7 $\mu$g. . .

Page 5, line 25, Results: something is wrong with this sentence, revise

Page 6, line 23: change "weres" to "were"

---

## Referee Comment (RC2) · Anonymous Referee #2 · 31 May 2019

**Review of: "Interferences of volatile organic compounds (VOC) on methane concentration measurements" by Kohl et al.**

The paper studies experimentally the interferences of several VOCs on the measurement results of several $CH_4$ analyzers. VOCs interfere strongly with FTIR but not with laser absorption spectroscopy measurements of $CH_4$. The results indicate that the FTIR instruments are not suitable for $CH_4$ measurements in high-VOC conditions, e.g. when estimating $CH_4$ fluxes from plants or soil. Laser absorption spectrometers are much less affected by VOC interference, thus can be used in high-VOC conditions. Including the main VOCs in the FTIR library corrects for part but not all the interference on methane. A by-product of this study is the finding that VOCs can be quantified by FTIR, at least at the high concentrations used here.

The paper is very useful given the recent increase in attention to $CH_4$ emissions from or via trees, and the increasing availability of field capable instruments.

The paper is well written and I recommend publication after the comments below are addressed.

**General comments**

- I think it is important to discuss the relevance of these findings for the recent studies of methane emissions form trees (e.g. summarized in Covey et al., 2019). Did any of these studies use FTIR instruments?
- "Concentration" is not the correct term for molar ratios (i.e. all the quantities expressed as ppm or ppb). "Mole fraction" or "mixing ratio" should be used instead.
- An explanation is missing on how the VOCs to be tested were chosen. Are these representative for real world emissions from vegetation?
- The VOC concentrations used in the lab experiments seem quite high. Are these representative for what one can expect in a tree chamber? Consider mentioning this in the method already.
  Also, when discussing the sensitivies of $CH_4$ to VOCs, it would be useful to relate to real world expected VOC levels.
- not all VOCs from Test 1 were used in Test 2 – why? Did the ones that were removed not have an influence? Especially alpha-pinene, which the authors mention it is the main VOC emitted by spruce.
- two different FTIR instruments were used, one in the field campaign and Test 2, and the other one in Test 1. Are these similar enough that the results can be considered together? If yes, please state in the text. Otherwise they should probably be treated separately through the paper.

**Specific comments**

- at the end of Introduction the authors state that the test setup was built. I suggest adding one sentence stating clearly what is presented in this paper: the field experiments? or the lab test setup? the results of both?
- page 2 lines 14-19: the phrase is a bit long and hard to follow, with some commas missing. Please consider reformulating.

- page 3 lines 6-7: specify what the inhouse pressured air supply is based on: e.g. gas cylinder(s) or a large compressor taking outside air. This is relevant for how the uncertainty is calculated (page 4, and see comment below)
- page 3 line 21 and page 4 line 9: are δ3-carene and Δ3 -carene the same chemical?
- Fig. 3: Caption – specify the experiment these data come from. For panel *a*, the text says "development of VOC concentration" but only beta-pinene is shown.
- page 4 lines 22-30: if the inhouse supply of pressured air takes atmospheric air from outside, there will be non-random variations on diurnal time scales, with e.g. possibly large methane increase during night. Is this taken into account in the bootstrap, i.e. are the 500 time intervals from the same part of day as the VOC experiments? Or was the day/night variation in the inhouse air estimated?
- Fig. 4: I find some parts of Fig. 4 confusing. In panels *a* and *b* it is not easy to understand which trace corresponds to which y-axis. E.g., in the upper middle panel, do the the methane data correspond to the blue unlabeled scale, or to the side scales labelled "$CH_4$"? What does the blue y-axix represent, and what are the units? Consider splitting the panels. Similar for panels *g*, *h*, *i*. Also, please consider splitting Fig 4 into two figures.
- page 5 Sect 3.1: suggest to refer to Fig 1.
- page 7 line 13: was alpha-pinene not included in Test 1?

**Text comments:**

- page 1, line 7: typo "strong strong"
- page 2 line 29: typo "Summer"
- page 3 line 6: "Fig 3a" – should it also be "Fig 2a", since this is the setup decription?
- page 3 lines 9-10: "The flow air" – should it be "the air"?
- page 3 line 29: "measure of VOC interferences" should be "measure the VOC interferences"?
- page 3 line 30: "Fig 3b" – should it be "Fig 2b"?
- page 4 line 25: "those by a random period" should be "by those from a random period" – please check
- page 4 line 29: "Significance interference" should probably be "Significant interference"
- page 4 line 31: "to evaluate"
- page 4 line 32: "we evaluated calculating" should be "we calculated" ?
- page 5 line 26: probably typo: [spikes?]
- page 6 line 10: typo "/beta"
- page 6 line 14: I think "and not part of …" should be "was not part of …"
- page 6 line 15: ")" missing after "VOCs"

---

## Author Comment (AC3) · 29 Jul 2019

[revised manuscript text omitted]

ANDSCAPE TABLE

---

## Author Response (AR1)

Dear Dr. Niemann, Dear Reviewers,

Please find attached our revised manuscript "Technical note: Interferences of volatile organic compounds (VOC) on methane concentration measurements". We thank the two reviewers for their constructive feedback, which has helped to further improve the manuscript. Please see below our detailed response to each of the reviewers' comments.

Both reviewers suggested that the manuscript should clarify whether the VOC mixing ratios applied in our experiment are representative for actual chamber measurements. In the revised manuscript, we now present estimates for the VOC mixing ratios expected at the end of soil, stem, and shoot chamber closures along with estimates of the bias VOC interferences excert over CH4 flux measurements in these chambers. While these estimates rely heavily on assumptions and simplifications, we hope that they provide the reader with a better understanding for where relevant VOC interferences are to be expected and what order of magnitude they can reach in different ecosystem compartments.

All other comments were addressed to meet the reviewers' recommendations. Given the short length of the technical note, we were unable to incorporate answers to all of the reviewers' technical questions in the manuscript itself, instead we provide answers to some of the reviewers' questions this (public) response letter.

Yours sincerely,

Lukas Kohl (on behalf of all co-authors)

Detailed response to reviewer comments

(*reviewer comments in italic,* our response in normal font. We abbreviate page and line numbers such that p2 L15 refers to page 2 line 15. We apologize for inconveniences caused by line numbers restarting with every new page which, unfortunately, is set by the Biogeosciences LaTeX template)

Editor's comments:

*Dear Lukas Kohl and co-authors,*

*two anonymous reviewers evaluated your MS and both seem quite positive about your work. I found your replies good, too and would this like to prepare a revised version of your MS for consideration after minor revisions. Please note that a new MS file needs to be uploaded (I noted that you uploaded your revised MS as an author comment in the discussion, and thought the revised version seems fine for the most part, note that it needs to be uploaded separately).*

*In addition to the reviewer comments, I would like you to clarify in the MS if differential material is used (ie standard alpha pinene).*

I assume that this refers our response to R2.5 ("Unfortunately, we ran out of our α-pinene standard during experiment 2 and therefore used β-pinene and Δ3-carene to represent monoterpenes").

For clarification, we tested three monoterpenes in experiment 1 (α-pinene β-pinene, and Δ3-carene), and two monoterpenes in experiment 2 (β-pinene and Δ3-carene). In our response we explain why we chose these two monoterpenes for experiment 2 and left out α-pinene (we ran out of the standard). We did not want to imply that any standard was changed between the experiments. This should be clear from p3 L29-30 and p4 L18-19.

*Also make sure that figures are well readable. While I found fig 3 easy to interpret, figs 4 and 5 are composed of rather thin lines , the colour scheme of which gets difficult to see, particularly if these are furhter shrunk.*

We increased the line width and legend font size and added different line weights to make it easier to discriminate between the lines. I hope the figure is easier to read now – please let us know if further changes to this figure are required.

*It is also not clear to me why the apparent methane concentrations of the different instruments are plotted on different scales.*

The different scales are due to the vast differences in instrument precision and detected interferences. For the LGR and Picarro instruments, we want to highlight that the measured $CH_4$ mixing ratios are constant with very high precision (on the scale of single ppb). For FTIR with the limited library, we want to show that the measured $CH_4$ mixing ratios vary on the scale of ppm, and that these variations follow the same pattern as the β-pinene concentrations. Finally, we want to show that these interferences are minimized when FTIR data is analysed with a complete spectral library.

Reviewer 1:

*General comments*

*The paper by Kohl et al. describes cross sensitivities of several volatile organic compounds on methane measurements when using different optical analysers. I consider the results of the paper of major interest to all those monitoring methane fluxes in the field or laboratory from ecosystems and biological systems that are known to release VOC at substantial amounts. I found the manuscript to be well written and structured.The results are clearly presented and discussed in a straightforward manner, providing the scientific community with important information about how emissions of VOC released from the biosphere might interfere with measurements of methane when using state of the art optical measurement systems. I recommend publication of the manuscript as a Technical Note in Biogeosciences after minor revisions. I have only a few comments which I hope the authors might consider in their revised manuscript.*

R1.1: *I would suggest using ppmv/ppbv/pptv (parts per million/billion/trillion by volume) throughout the whole manuscript instead of ppm/ppb/ppt.*

**Changed throughout the manuscript.**

R1.2 *Furthermore, the correct expression for ppmv would be mole fraction. However, I also understand if the authors would like to keep the more commonly used term "concentration".*

**Changed to 'mixing ratio' throughout the manuscript.** We kept the more commonly used term 'concentration' in the title.

R1.3 *As water vapour might substantially affect measurements of methane (both concentrations and stable carbon isotopes) when using optical analyzers I would suggest to add a few sentences how the authors have dealt with this issue during their investigations in the field and in the laboratory.*

Laboratory measurements: Water was removed from the pressurized air used for the laboratory experiments (SMC membrane dryier) and water contents remained <0.2% absolute humidity throughout the experiment. Water vapour therefore did not affect $CH_4$ concentration or stable carbon isotope measurements.

Field measurements: Both analysers quantified water concentrations and used these concentrations to corrected $CH_4$ concentrations. No carbon isotope values were measured during the field measurements reported in this manuscript.

R1.4 *Please add some information what are typical emission rates of some VOC released from vegetation/trees in the field and put them into relation with the amounts that have been applied in the laboratory study.*

**Changed as requested.** Thanks for this suggestion; we think that this adding such information strengthened the paper a lot. Typical VOC emission rates and estimates for mixing ratios reached during chamber closures are now provided in the new Tables 1 and 4. Overall, the mixing ratios employed in our experiment are above those likely to occur in soils and stem chambers, but below those likely found in shoot chambers.

R1.5 *Figure 4: There are too many subfigures included and for some subfigures it is rather difficult to decipher the information. Please revise and split into two or three figures to increase readability.*

**Changed as requested.** We removed three panels and split Fig. 4 two figures (new Figs. 4 and 5)

*Technical corrections*

R1.6 *Page 5, line 6, Results: add CH4 after 7μg...*

**Changed** (p5 L9).

R1.7 *Page 5, line 25, Results: something is wrong with this sentence, revise*

**Changed** (p5 L29).

R1.8 *Page 6, line 23: change "weres" to "were"*

**Changed** (p6 L29).

Reviewer 2

*Review of: "Interferences of volatile organic compounds (VOC )on methane concentration measurements" by Kohl et al.*

*The paper studies experimentally the interferences of several VOCs on the measurement results of several CH4 analysers. VOCs interfere strongly with FTIR but not with laser absorption spectroscopy measurements of CH4. The results indicate that the FTIR instruments are not suitable for CH4 measurements in high-VOC conditions, e.g. when estimating CH4 fluxes from plants or soil. Laser absorption spectrometers are much less affected by VOC interference, thus can be used in high-VOC conditions. Including the main VOCs in the FTIR library corrects for part but not all the interference on methane. A by-product of this study is the finding that VOCs can be quantified by FTIR, at least at the high concentrations used here.*

*The paper is very useful given the recent increase in attention to CH4 emissions from or via trees, and the increasing availability of field capable instruments. The paper is well written and I recommend publication after the comments below are addressed.*

*General comments*

R2.1 *I think it is important to discuss the relevance of these findings for the recent studies of methane emissions form trees (e.g. summarized in Covey et al., 2019). Did any of these studies use FTIR instruments?*

> We are unaware of any published tree $CH_4$ flux data that used FTIR based instruments. Many of the studies summarized by Covey et al us gas chromatography to quantify $CH_4$ (which is not vulnerable to the interferences described herein), while some of the more recent studies quantified $CH_4$ by laser spectroscopy (Picarro and LGR instruments). We are, however, aware of several groups currently considering the use of FTIR instruments for stem flux measurements. We therefore think that the reliability of currently available data is not impacted by our work, but that this manuscript is important as the potential use of FTIR for tree stem flux measurements would decrease this data reliability in the future.

R2.2 *"Concentration" is not the correct term for molar ratios (i.e. all the quantities expressed as ppm or ppb). "Mole fraction" or "mixing ratio" should be used instead.*

> **Changed to 'mixing ratio' throughout the manuscript.** We kept the more commonly used term 'concentration' in the title.

R2.3 *An explanation is missing on how the VOCs to be tested were chosen. Are these representative for real world emissions from vegetation?*

**Clarified as requested.** p3 L3-5 now read "We chose the tested compounds to represent a cross-section of naturally occurring VOCs and aimed to cover different chemical compound classes rather than the most important biogenic VOCs occurring in any given environment."

R2.4 *The VOC concentrations used in the lab experiments seem quite high. Are these representative for what one can expect in a tree chamber? Consider mentioning this in the method already. Also, when discussing the sensitivities of CH4 to VOCs, it would be useful to relate to real world expected VOC levels.*

**See response to R1.4.** We added the new Sections 2.5 and 3.5, Fig. 7, and Tables 1 and 4 to provide estimates for VOC mixing ratios reached during chamber closures.

R2.5 *not all VOCs from Test 1 were used in Test 2 –why? Did the ones that were removed not have an influence? Especially alpha-pinene, which the authors mention it is the main VOC emitted by spruce.*

Due to time constraints and limited instrument availability. While the tests conducted during Experiment 1 took around 1h per compound, tests in Experiment 2 took one overnight run per compound. We chose the VOCs tested to cover a broad diversity of chemical compound classes (monmoterpenes, methanol, aliphatic and aromatic compounds). Unfortunately, we ran out of our alpha-pinene standard during experiment 2 and therefore used β-pinene and Δ3-carene to represent monoterpenes.

R2.6 *two different FTIR instruments were used, one in the field campaign and Test 2, and the other one in Test 1. Are these similar enough that the results can be considered together? If yes, please state in the text. Otherwise they should probably be treated separately through the paper.*

**Clarified as requested.** These are very similar instruments (DX4040 is the portable version of DX4015). They have the same measurement cell, detector technology, and spectral deconvolution software. p4 L16-18 now read "[...] we replaced the FTIR-based analyser with a portable but otherwise similar model [...]"

*Specific comments*

R2.7 *at the end of Introduction the authors state that the test setup was built. I suggest adding one sentence stating clearly what is presented in this paper: the field experiments? or the lab test setup? the results of both?*

**Modified as requested.** We added the following sentence at the end of the introduction "In this communication, we present results from field measurements and laboratory tests, as well as a first sensitivity analysis for the impact of VOC interferences on measurements of $CH_4$ fluxes from different ecosystem compartments." (p2 L28-30)

R2.8 *page 2 lines 14-19: the phrase is a bit long and hard to follow, with some commas missing. Please consider reformulating.*

**Modified as requested.** p2 L19-21 now read "This is especially important in the study $CH_4$ emissions by plants as plants co-emit a complex mixture of volatile organic compounds (VOC) at fluxes 2 to 4 orders of magnitude higher than currently reported $CH_4$ fluxes [references]."

R2.9 *page 3 lines 6-7: specify what the in house pressured air supply is based on: e.g. gas cylinder(s) or a large compressor taking outside air. This is relevant for how the uncertainty is calculated (page 4, and see comment below)*

**Clarified as requested.** The air was taken from a compressor using outside air. (p3 L18)

R2.10 *page 3 line 21 and page 4 line 9: are δ3-carene and Δ3 -carene the same chemical?*

**Corrected.** This should be a uppercase delta in all cases. (p4 L23)

R2.11 *Fig. 3: Caption –specify the experiment these data come from. For panel a, the text says "development of VOC concentration" but only beta-pinene is shown.*

**Changed as requested.** The caption to Fig 3. now starts "Exemplary results from Experiment 1, shown for tests conducted with β-pinene."

R2.12 *page 4 lines 22-30: if the in house supply of pressured air takes atmospheric air from outside, there will be non-random variations on diurnal time scales, with e.g. possibly large methane increase during night. Is this taken into account in the bootstrap, i.e. are the 500 time intervals from the same part of day as the VOC experiments? Or was the day/night variation in the inhouse air estimated?*

The data used for bootstrapping was collected during nighttime (7pm to 7am). Experiment 2 runs were started between 10am and 4pm and ran until 1am to 8am. This means that there is indeed a small potential that we underestimated non-random variations in CH4 concentrations that occurred during daytime. This affects mainly gradient challenges, which were conducted before the stepwise challanges in the same run.

The bootstrapping approach was employed to account for the added uncertainty due to drifts in the inlet $CH_4$ mixing ratio. These additional uncertainties were largely symmetrical, which suggests that periods of increasing and decreasing $CH_4$ concentrations were equally represented in the data used for bootstrapping. We conducted every individual challenge (VOC / analysers / stepwise-or-gradient combination) at least twice, with >1.5h (gradients) or >4h (stepwise) between measurement. Overall, we think that in spite of diurnal variations estimates still represent a fairly conservative estimate for the true uncertainty in our experiments.

R2.13 *Fig. 4: I find some parts of Fig. 4 confusing. In panels a and b it is not easy to understand which trace corresponds to which y-axis. E.g., in the upper middle panel, do the methane data correspond to the blue unlabeled scale, or to the side scales labelled "CH4"? What does the blue y-*

*axix represent, and what are the units? Consider splitting the panels. Similar for panels g, h, i. Also, please consider splitting Fig 4 into two figures.*

> **Changed as requested.** We split Fig 4 into two separate figures (new Figs. 4 and 5), removed three panels, and revised the corresponding figure captions.

R2.14 *page 5 Sect 3.1: suggest to refer to Fig 1.*

> **Changed as requested.** We moved the reference to Fig 1 up by one sentence to meet the first mention of data from Fig. 1 in this paragraph (p6 L9).

R2.15 *page 7 line 13: was alpha-pinene not included in Test 1?*

> No. While we did screened for (and detected) interferences by α-pinene in experiment 1, but we did not conduct quantitative measurements of α-pinene interferences (hence, we note that they were not quantified.)

*Text comments:*

R2.16 *page 1, line 7: typo "strong strong"*

> **Corrected** (p1 L9).

R2.17 *page 2 line 29: typo "Summer"*

> **Corrected.** (p2 L4).

R2.18 *page 3 line 6: "Fig 3a" –should it also be "Fig 2a", since this is the setup decription?*

> **Corrected** (p3 L17).

R2.19 *page 3 lines 9-10: "The flowair" –should it be "the air"?*

> **Corrected** (p3 L21).

R2.20 *page 3 line 29: "measure of VOC interferences" should be "measure the VOC interferences"?*

> **Changed to "measure VOC interferences".** (p4 L10)

R2.21 *page 3 line 30: "Fig 3b" –should it be "Fig 2b"?*

> **Corrected** (p4 L11).

R2.22 *page 4 line 25: "those by a random period" should be "by those from a random period"*

> **Corrected** (p5 L10).

R2.23 *please check-page 4 line 29: "Significance interference" should probably be "Significant interference"*

**Corrected** (p5 L 14).

R2.24 *page 4 line 31: "to evaluate"-page 4 line 32: "we evaluated calculating" should be "we calculated" ?*

**Corrected** (p5 L17).

R2.25 *page 5 line 26: probably typo: [spikes?]*

**Corrected to 'outliers'.** (p6 L30)

R2.26 *page 6 line 10: typo "/beta"*

**Corrected** (p7 L15).

R2.27 *page 6 line 14: I think "and not part of ..." should be "was not part of ..."*

**Corrected** (p7 L 19).

R2.28 *page 6 line 15: ")" missing after "VOCs"*

**Corrected** (p7 L20).

[revised manuscript text omitted]